# Transcriptome-Wide Identification of Cytochrome P450s and GSTs from *Spodoptera exigua* Reveals Candidate Genes Involved in Camptothecin Detoxification

Zhenzhen Zhao, Lan Zhang *, Yanning Zhang, Liangang Mao, Lizhen Zhu, Xingang Liu and Hongyun Jiang

State Key Laboratory for Biology of Plant Diseases and Insect Pests, Institute of Plant Protection, Chinese Academy of Agricultural Sciences, Beijing 100193, China; zhulizhen@caas.cn (L.Z.)
* Correspondence: lanzhang@ippcaas.cn

**Abstract:** The application of camptothecin (CPT) and its derivatives to control insect pests has generated significant interest. This study investigated the enzymatic response of cytochrome P450 monooxygenase (CYP) and glutathione S-transferase (GST) genes in the fat body cells of *Spodoptera exigua* after 10 μM CPT treatment. Additionally, we examined the effects of CPT on the growth and development of *S. exigua* larvae and detected the relative expression levels of selected CYP and GST genes during the CPT treatment period. Twenty-one CYP and 17 GST genes were identified from the fat body cells of *S. exigua* by comparative transcriptomic analyses. The relative expression of ten CYP and seven GST genes changed significantly, which suggested that these CPYs and GSTs may be involved in CPT metabolism. During exposure to CPT for 10 days, the development of *S. exigua* larvae was delayed and was characterized by weight inhibition and a prolonged period of development. The relative expression levels of the selected four CYP genes, *CYP9A27*, *CYP9A186*, *CYP337B5*, *CYP321A8*, and one GST gene, *GSTe7*, were significantly changed by CPT treatment compared to the control group. These generated data provide a basis for identifying the CPT metabolism/detoxification genes of *S. exigua* at the molecular level.

**Keywords:** cytochrome P450 monooxygenase; glutathione S-transferases; camptothecin; *Spodoptera exigua*

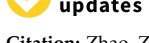



## 1. Introduction

Camptothecin (CPT), a botanical indole alkaloid, was isolated from *Campototheca acuminata*, which is native to China. CPT and its derivatives (CPTs) are well known for their antitumor activity and are considered to be an important class of drugs used in chemotherapy. It has been documented that its complex metabolism involves many proteins. In humans, cytochrome P450 monooxygenases (P450s or CYPs) have been documented as being involved in the degradation of CPTs, especially P450 isoforms 1A1, 3A4, and 3A5 (*CYP1A1*, *CYP3A4*, *CYP3A5*) [1–3]. In recent years, based on the knowledge of the metabolism of CPTs by CYPs, a promising CPT delivery scaffold—CPT-encapsulated PLGA (poly(lactic-co-glycolic acid), PLGA-CPT) nanoparticles—was prepared to improve the bioavailability of CPTs and reduce their side effects [4]. These studies highlight the importance of the metabolism of CPTs for drug discovery and predicting the toxicity and efficacy of CPTs in drug therapy [5].

In agriculture, much interest has been focused on developing CPTs as potential biological pesticides to control agricultural pests due to the unique action mechanisms and insecticidal activities of CPTs. It has been documented that CPTs are effective against several pest insects, including *Spodoptera exigua* [6,7], *S. litura* [8], *S. frugiperda* [9], *Nilaparvata lugens*, *Brevicoryne brassicae*, and *Chilo suppressalis* [10]. In our previous studies, CPTs induced cytotoxicity against IOZCAS-Spex-II cells derived from *S. exigua* by promoting apoptosis [11] and increasing intracellular oxidative stress [12,13]. In addition, a series of CPT derivatives

was designed and synthesized to increase the solubility and bioavailability of CPT [6,14–16]. With the development of nanopesticides, CPT was conjugated with polyethylene glycol by using nanotechnology to form an amphiphilic copolymer, mPEG-CPT, which exhibited potent or superior insecticidal activities against three pests, *B. brassicae*, *Tetranychus cinnabarinus*, and *Bursaphelenchus xylophilus* [17]. However, to date, little is known about the metabolism of CPTs in insects. To improve the bioavailability of CPTs and promote the application of CPTs in the control of insect pests, the metabolism of CPTs should be investigated comprehensively in insects.

The detoxification systems of insects have been shown to play central roles in the metabolism of a wide range of natural and synthetic xenobiotics, including insecticides. The most frequently involved enzyme families contributing to the metabolism of xenobiotics included CYPs, glutathione S-transferases (GSTs), and carboxylesterase (CarEs), although other enzyme families may also be involved [18]. For insecticides, gene amplification, overexpression and/or the modification of genes encoding members of the CYPs and GSTs must be involved in the metabolic detoxification of multiple insecticides [19,20]. In the present study, we investigated the enzymatic response, including CYP and GST genes, in the fat body cells of *S. exigua* after 10 µM CPT treatment. Furthermore, the effects of CPT on the growth and development of *S. exigua* larvae were examined, and the relative expression levels of the selected CYP and GST genes were monitored during the CPT treatment period.

## 2. Materials and Methods

### 2.1. Chemicals, Cell Line, and Culture Conditions

CPT (98%) was supplied by Knowshine Pharmachemicals Inc., Shanghai, China. Dimethyl sulfoxide (DMSO, $\geq$99.5%) was purchased from Sigma Aldrich Co., Shanghai, China.

### 2.2. CPT Treatment of S. exigua Fat Body Cells

IOZCAS-Spex-II cells were established from the fat body of *S. exigua* and offered by Prof. Qilian Qing of the Institute of Zoology, Chinese Academy of Sciences, Beijing, China. The cells were grown in T25 cm$^2$ tissue culture flasks (Corning, New York, NY, USA) and maintained in Grace's insect culture medium supplemented with 10% fetal bovine serum. The culture was incubated at a temperature of 27 °C. Subculturing was performed every 6 days.

IOZCAS-Spex-II cells in the logarithmic phase were harvested and diluted to a density of $2 \times 10^6$ cells mL$^{-1}$ in the cell cultures. Following overnight incubation, the cells were exposed to CPT (10 µM well$^{-1}$) at appropriate concentrations for 24 h. DMSO (0.1%) was used as a control. Each treatment was repeated at least three times.

### 2.3. RNA Extraction, Library Construction and Sequencing

Total RNA was extracted from cell samples using the RNeasy Mini Kit (Qiagen, Hilden, Germany) according to the manufacturer's instructions. The integrity of the total RNA was confirmed by using both 1.5% agarose gel electrophoresis and an Agilent 2100 Bioanalyzer (Agilent Technologies, Santa Clara, CA, USA). The concentration of the total RNA was determined using a NanoDrop 1000 spectrophotometer (Thermo Fisher Scientific Inc. Waltham, MA, USA). cDNA libraries were constructed using an Illumina TruSeq RNA sample preparation kit following the manufacturer's instructions (Illumina Inc., San Diego, CA, USA). After quantification and qualification, the cDNA libraries were sequenced with Illumina HiSeqTM 2000 in accordance with the standard procedure.

The raw reads were filtered to remove reads containing adapter, poly-N, and low-quality reads. The resulting clean reads were subjected to pool de novo transcriptome assembly using the Trinity-v2.3.2 assembler programs at default parameters without a reference genome. All de novo transcripts were predicted to have open reading frames (ORFs) with TransDecoder software (v5.5.0) and then corrected with the Pfam database. ORF annotation was performed against the NCBI nonredundant protein database (NR)

and SwissProt database. The gene expression level was computed using RSEM quantitative analysis and FPKM transformation. edgeR-2020 software was used to screen the differentially expressed genes between the two samples with a fold change (FC) $\geq$ 1 and false discovery rate (FDR) < 0.01 as the threshold by which to judge the significance of differences in gene expression.

### 2.4. Identification of Detoxification Genes and Phylogenetic Analysis

The assembled transcripts associated with CYPs and GSTs were manually selected and identified from the transcriptome data of the control and CPT-treated groups. These candidate sequences were realigned by BLAST in NCBI to find homology, and preliminary classification was based on the comparison results (Table S1).

The full protein sequences of all putative detoxification enzymes were obtained and analyzed using the ExPASy (the Expert Protein Analysis System) server http://web.expasy.org/translate/ (accessed on 11 November 2022). Due to the size differences of the two detoxification gene repertoires, the corresponding protein sequences were selected from different species to build the trees (Table S2). To assign predicted *S. exigua* detoxification genes to different phylogenetic subgroups, multiple sequence alignments of the CYP and GST amino acids were performed using MUSCLE in MEGA 11 software. The phylogenetic tree was generated using the neighbor-joining method of the MEGA 11 program with a bootstrap value of 1000. The webserver https://www.evolgenius.info/evolview (accessed on 5 December 2022), Evolview v3, was used for the visualization, annotation, and management of phylogenetic trees [21].

### 2.5. Real-Time Quantitative PCR (RT-qPCR)

RT-qPCR was performed to verify the expression of candidate CYP and GST genes. The PCR primer sequences used for quantification of the 43 candidate genes are shown in Table S3. GAPDH (glyceraldehyde-3-phosphate dehydrogenase) and L7A (ribosomal protein L7A) were used as the reference genes in this study [22]. A standard curve for each gene was established to evaluate the reaction efficiency of RT-qPCR. The melt curves were examined to check the specificity of the primers. RT-qPCR was carried out using the Hieff UNICON® Universal Blue qPCR SYBR Green Master Mix (YEASEN, Shanghai, China) on an Applied Biosystems QuantStudio 3 Real-Time PCR System (ABI, Carlsbad, CA, USA). Each 20 μL reaction contained 10 μL SYBR Green, 0.6 μL of each primer, 1 μL cDNA template, and double distilled water. The cycling parameters consisted of an initial step at 95 °C for 120 s, followed by 40 cycles of 95 °C for 10 s and 55–60 °C for 30 s.

### 2.6. Effect of CPT on the Growth and Development of S. exigua Larvae

A stock solution of $1.00 \times 10^5$ mg/L CPT was prepared with DMSO, which was then diluted with dH$_2$O (containing 0.05% Triton X-100) to prepare the tested solution. Uniform second-instar larvae were reared on cabbage leaves dipped in CPT at concentrations of 0.02, 0.1, and 0.5 g/L for 10 days. Fresh cabbage dipped in CPT was provided every day and five replicates were made for each treatment with 20 individuals per replicate. Larvae were reared on cabbage leaves treated with dH$_2$O (containing 0.05% Triton X-100 and 0.1% DMSO) served as the control group. The survival rate, weight, and developmental period of each individual were checked daily. The rearing conditions for *S. exigua* larvae were $27 \pm 1$ °C and $60 \pm 5$% relative humidity (RH) with a photoperiod of 14 h of light followed by 10 h of darkness 14:10 h (L:D).

### 2.7. Effect of CPT on the Expression of Detoxification Genes in S. exigua Larvae

Total RNA was extracted from *S. exigua* larvae with a TRNzol Universal total RNA extraction kit following the instructions provided by TIANgen Biotech Co., Ltd. Primary strand cDNA was generated from 1 μg of total RNA in a 20 μL volume using the Omniscript® Reverse Transcript Kit (Qiagen, ON, Canada) and was stored at −20 °C for future analysis. RT-qPCR was performed as described in Section 2.3.

## 3. Results

### 3.1. Identification of Candidate P450s and GSTs from the Fat Body Cells of S. exigua

As shown in Figure 1a, twenty-one putative CYP genes among the DEGs were identified from the transcriptome data obtained from the fat body cells of *S. exigua*. The phylogenetic tree resolved the four expected clans of insect CYPs, including CYP2, CYP3, CYP4, and the mitochondrial CYP clans. Specifically, thirteen genes were classified into the CYP3 clan, four into the CYP4 clan, three into the mitochondrial CYP clan, and one into the CYP2 clan. Additionally, the 21 CYPs belonged to 10 families and 17 subfamilies. The relative expression of nine genes, *CYP6AB14*, *CYP6AB31*, *CYP6AB60*, *CYP337B5*, *CYP6AW1*, *CYP9A27*, *CYP9A186*, *CYP4G74* and *CYP333A12*, was upregulated significantly and one gene, *CYP333B4*, was downregulated significantly in *S. exigua* after treatment with 10 μM CPT for 24 h compared to the control group. The relative expression of a total of 11 genes, *CYP6AN4*, *CYP6AE74*, *CYP321A8*, *CYP354A14*, *CYP9AJ1*, *CYP324A1*, *CYP4M14*, *CYP4CG1*, *CYP4G75*, *CYP306A1*, and *CYP339A1*, was not changed in *S. exigua* after treatment with CPT compared with the control.

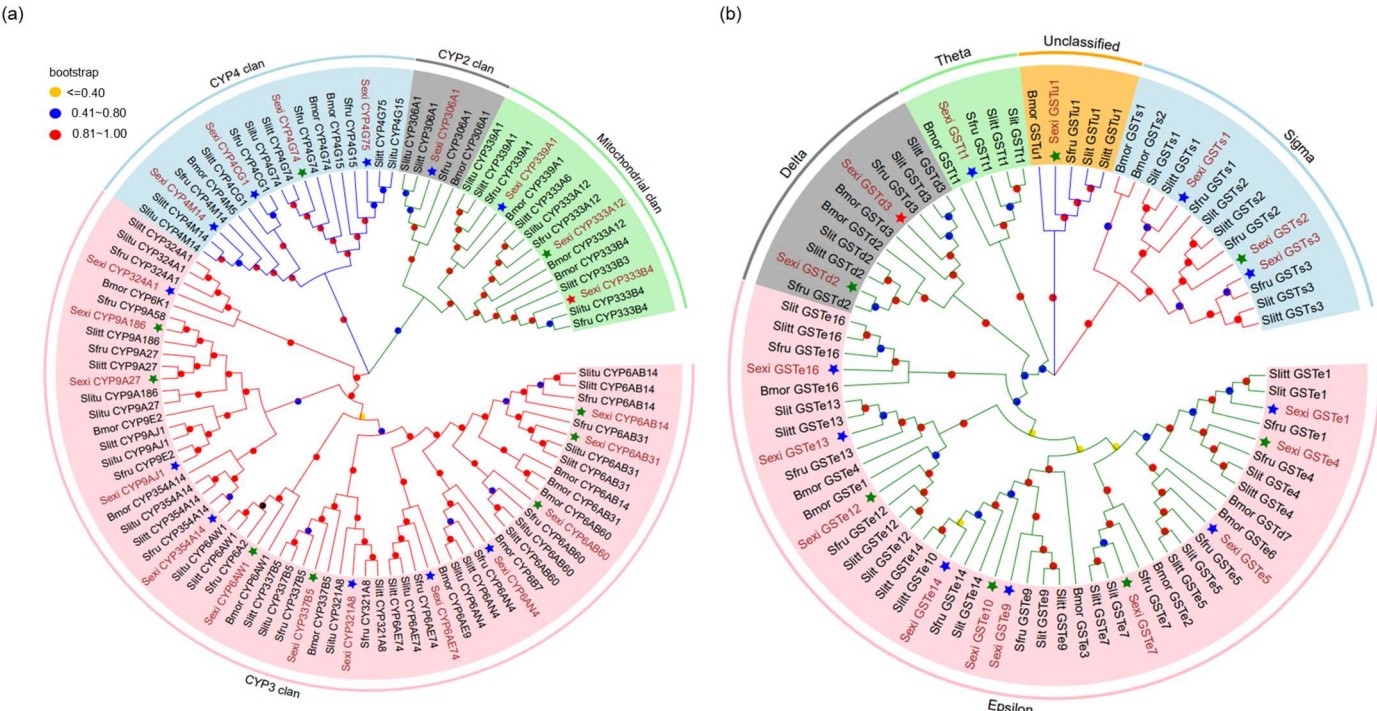

**Figure 1.** Phylogenetic analysis of selected CYPs (**a**) and GSTs (**b**) from *S. exigua* (Sexi), *S. litura* (Slitu), *S. littoralis* (Slitt), *S. frugiperda* (Sfru) and *Bombyx mori* (Bmor). The CYPs and GSTs of *S. exigua* are indicated in brown. The stars with different colors marked before CYP or GST proteins of *S. exigua* were significantly upregulated (green stars), significantly downregulated (red stars) or unchanged (blue stars) according to the transcriptome data ($|\mathrm{Log_2\ FC}| \geq 1$, $p < 0.05$).

Seventeen GST genes were identified from the transcriptome data of *S. exigua*. Among these, phylogenetic analyses revealed that five GST classes were assigned, ten in epsilon, two in delta, one in theta, three in sigma, and one in unclassified (Figure 1b). The relative expression of seven genes, *GSTe4*, *GSTe7*, *GSTe10*, *GSTe12*, *GSTd2*, *GSTu1*, and *GSTs2*, was upregulated significantly in *S. exigua* after treatment with 10 μM CPT for 24 h compared to the control group. One gene, *GSTd3*, was significantly downregulated following CPT treatment in *S. exigua*.

*3.2. Real-Time Quantitative PCR Validation of Candidate Genes from the Fat Body Cells of S. exigua*

As shown in Figure 2, the expression patterns of the selected CYP (Figure 2a) and GST (Figure 2b) genes in the fat body cells of *S. exgua* changed significantly after 10 μM CPT treatment based on RT-qPCR analysis. The changes in gene expression levels based on RT-qPCR were largely consistent with the transcriptomic data. This result showed that these detoxification enzyme genes, especially CYPs, including *CYP337B5*, *CYP4G74*, *CYP6AB14*, *CYP6AB31*, *CYP6AB60*, *CYP9A186*, and *CYP9A27*, and GSTs, including *GSTd2*, *GSTe10*, *GSTe7*, *GSTe12*, *GSTe4*, and *GSTu1*, were potentially involved in the detoxification of CTP in *S. exigua*.

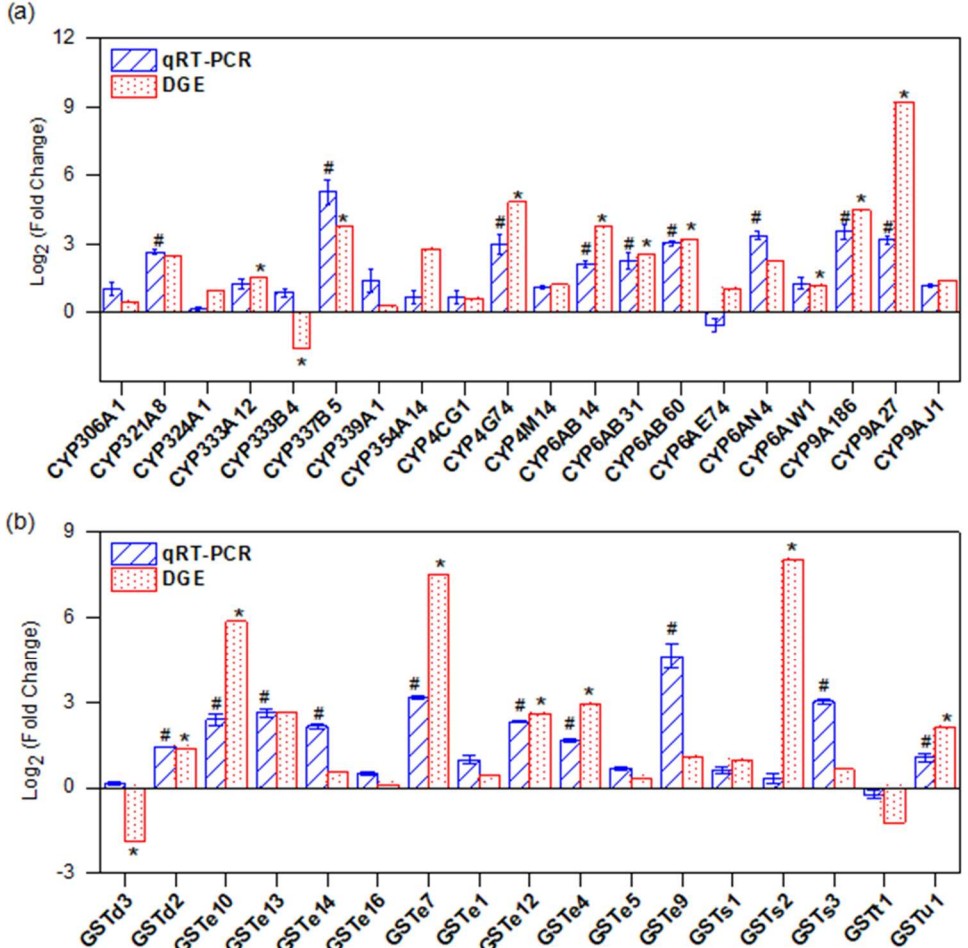

**Figure 2.** RT-qPCR validation of the expression of selected detoxification enzyme genes in *S. exigua* after CPT treatment. (**a**) CYP genes, (**b**) GST genes. # indicates significant differences ($|\text{Log}_2 \text{FC}| \geq 1$, $p < 0.05$) according to the transcriptome data. RT-qPCR experiments were confirmed in at least three independent experiments and the data are presented as the mean ± standard error. * indicates significant differences compared to the control group ($p < 0.05$). The SPSS 26.0 Software Package (SPSS Inc.; Chicago, IL, USA) was used to perform statistical analyses with Student's *t* test.

*3.3. Effects of CPT on the Development of S. exigua Larvae*

The survival rate, weight, and age-stage specific survival rate of *S. exigua* in different treatment groups are shown in Figures 3 and 4, respectively. The survival rates of *S. exigua* were 92.0%, 92.0%, and 71.0% after CPT treatment for 10 days at doses of 0.02, 0.1, and 0.5 g/L, respectively (Figure 3a). The survival rate of *S. exigua* larvae in the 0.5 g/L concentration group was significantly lower than that in the other two treatment groups after treatment with CPT for 10 days. Compared with the control group, CPT significantly

inhibited larval weight at all treatment doses. As shown in Figure 3b, the weight gain rate declined gradually from 82.7%, 68.6%, and 69.6% to 16.1%, 9.66%, and 5.54% after CPT treatment for 10 days at doses of 0.02, 0.1, and 0.5 g/L, respectively.

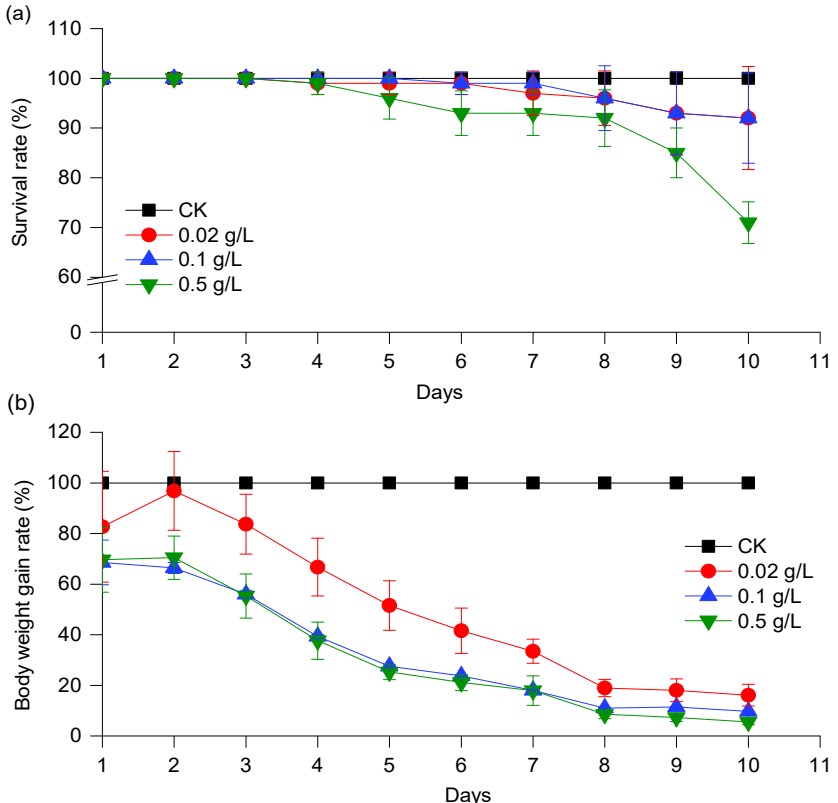

**Figure 3.** Effects of CPT on the survival (**a**) and body weight gain (**b**) of *S. exigua* larvae across 10 days of development. Data represent the mean ± SE.

The age-stage survival rate curves are shown in Figure 4 to illustrate the survival rates of newly hatched larvae developing to different stages. Observation overlaps were apparent between stages because the development rate varied among individuals. In the control group, the survival rates were 7.0%, 76.0%, and 17.0% for the 4th, 5th, and 6th instars at 10 days of development, respectively. Among the three CPT-treated groups, the survival rate decreased with increasing CPT concentration (1.0%, 82.0%, and 9.0% for the 3rd, 4th, and 5th instar larvae at a dose of 0.02 g/L, respectively; 22.0%, 67.0%, and 3.0% for the 3rd, 4th, and 5th instar larvae at a dose of 0.1 g/L, respectively; 57.0% and 14.0% for the 3rd and 4th instar larvae at a dose of 0.5 g/L, respectively). These results indicated that the larvae in the CPT-treated groups developed significantly later than those in the control group.

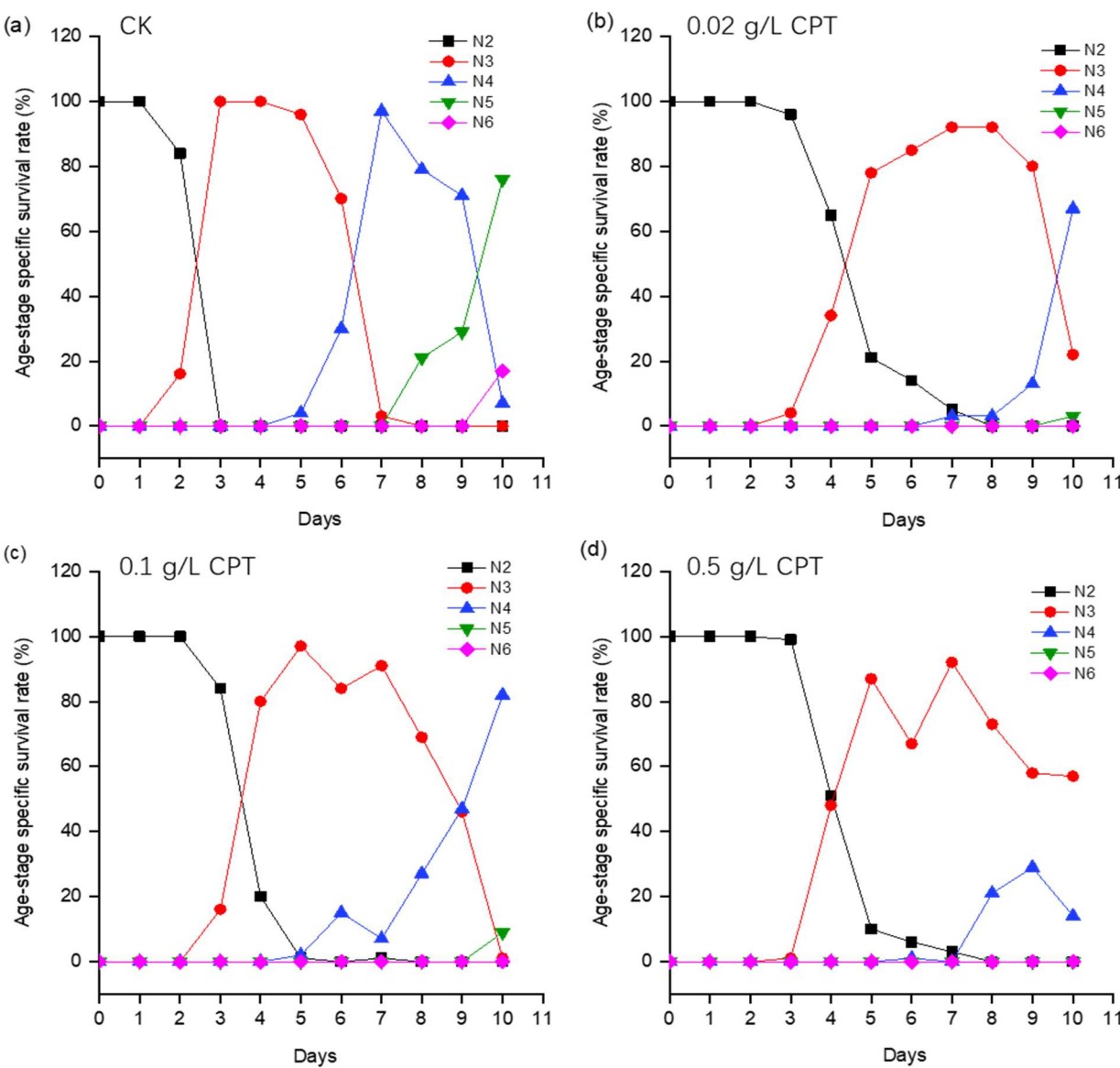

**Figure 4.** Age-stage-specific survival rate of *S. exigua* after exposure to CPT at different doses. N2–N6 are the larval instars from second to sixth. The age-stage-specific survival rate of the control group (**a**) and the groups treated with CPT at 0.02 g/L (**b**), 0.1 g/L (**c**), and 0.5 g/L (**d**).

*3.4. Effect of CPT on the Expression of CYP and GST Genes during the Development of S. exigua Larvae*

To study the effect of CPT on the expression of CYP and GST genes, four CYP3 clan genes (*CYP9A27*, *CYP9A186*, *CYP337B5*, and *CYP321A8*) and two GSTs (*GSTe8* and *GSTs3*) genes were selected to conduct further studies during the development of *S. exigua* larvae based on both results of transcriptome analysis and RT-qPCR verification. As shown in Figure 5, the expression of *CYP9A27*, *CYP9A186*, and *CYP337B5* varied with the development of *S. exigua* larvae in the control group (CK), which indicated that the *CYP9A27*, *CYP9A186*, and *CYP337B5* genes are potentially involved in the larval development of *S. exigua* (Figure 5a–c). However, the relative expression of *CYP321A8* remained unchanged throughout the tested period of *S. exigua* larval development in the control group (Figure 5d). Compared with those in the control group, the relative expression of *CYP9A27*, *CYP9A186*, *CYP337B5*, and *CYP321A8* was changed significantly by CPT treatment at different doses. Specifically, the relative expression of the *CYP9A27* gene was reduced significantly by CPT at all tested concentrations during the whole tested period.

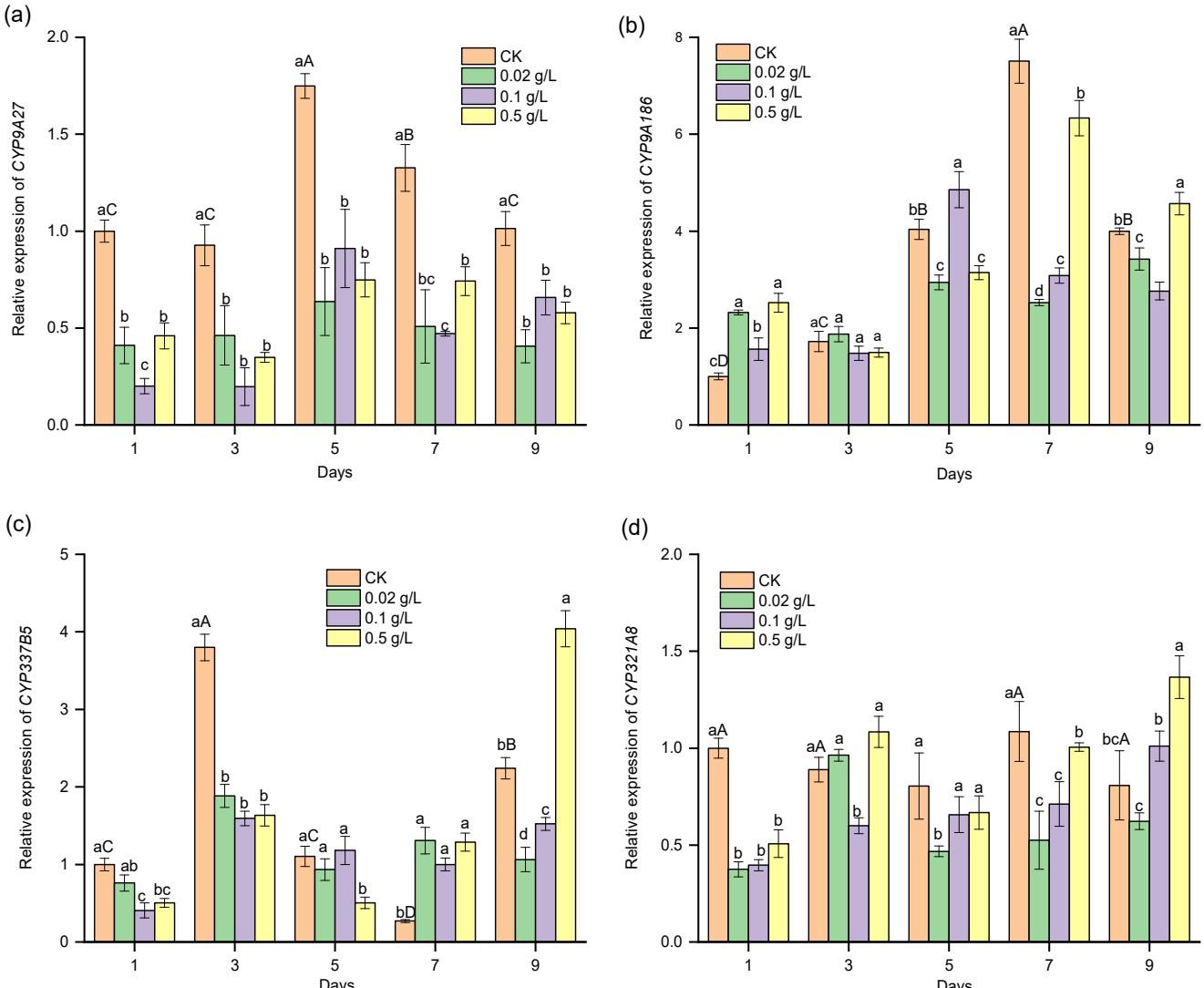

**Figure 5.** Effect of CPT on the expression of CYP genes during the development of *S. exigua*, including *CYP9A27* (**a**), *CYP9A186* (**b**), *CYP337B5* (**c**), and *CYP321A8* (**d**). RT-qPCR experiments were confirmed in at least three independent experiments and the data are presented as the mean ± standard error. Means followed by different small letters in the same time are significantly different ($p < 0.05$). To illustrate the changes in different CYP genes with the development of *S. exigua* larvae, the means of the relative expression in the control group were marked with different capital letters to indicate significant differences ($p < 0.05$). The SPSS 26.0 Software Package (SPSS Inc., Chicago, IL, USA) was used to perform statistical analyses by using Fisher's least significant difference (LSD) test.

As shown in Figure 6, the relative expression levels of the GST genes *GSTe7* and *GSTe9* were significantly reduced during the entire testing period in the control group. From 5 days to 9 days, the relative expression of *GSTe7* was increased significantly in the CPT-treated groups compared to the control group. The results show that *GSTe7* expression was significantly induced by CPT.

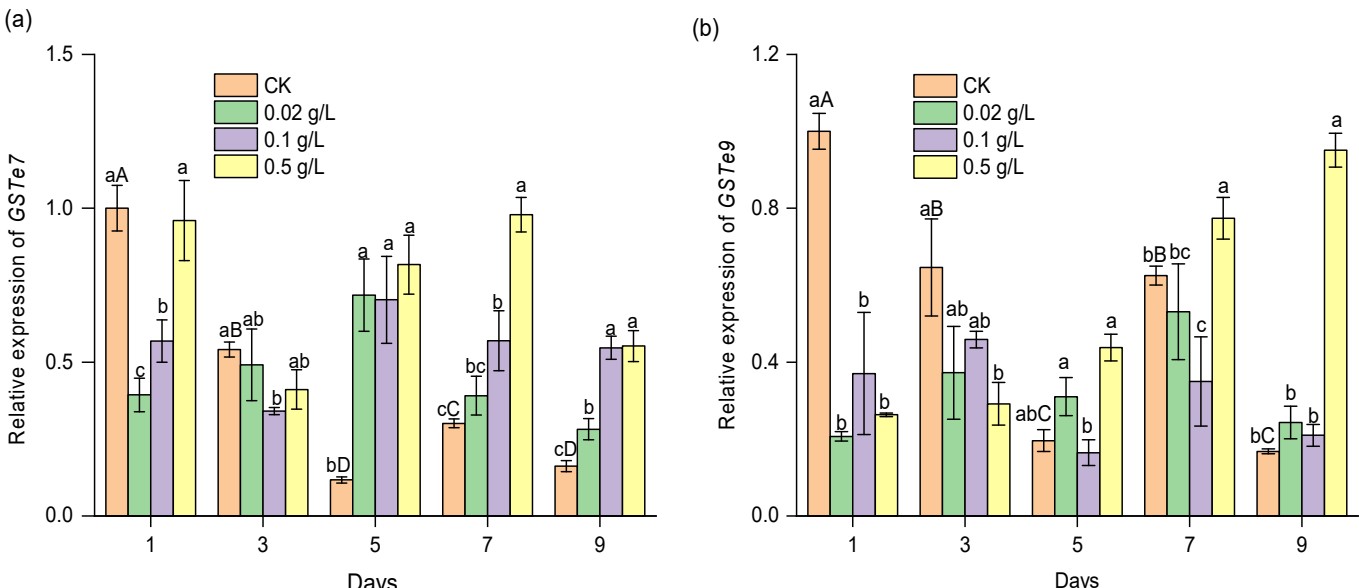

**Figure 6.** Effect of CPT on the expression of GST genes during the development of *S. exigua*, including *GSTe8* (**a**) and *GSTs3* (**b**). RT-qPCR experiments were confirmed in at least three independent experiments and the data are presented as the mean ± standard error. Means followed by different small letters in the same time are significantly different ($p < 0.05$). To illustrate the changes in different GST genes with the development of *S. exigua* larvae, the means of the relative expression in the control group were marked with different capital letters to indicate significant differences ($p < 0.05$). The SPSS 26.0 Software Package (SPSS Inc., Chicago, IL, USA) was used to perform statistical analyses by using Fisher's least significant difference (LSD) test.

## 4. Discussion

CYPs are a common and important metabolic detoxification gene family in insects. The fat body is a vital organ in insects that performs various functions, including xenobiotic metabolism and elimination. According to reports, a total of 53 CYPs were significantly differentially expressed with azadirachtin treatment in the fat bodies of *S. frugiperda* larvae, which indicated that these detoxification enzyme genes in fat bodies might be involved in azadirachtin detoxification in *S. frugiperda* [23]. In this study, a total of twenty-one putative CYP genes were identified from the transcriptomic data, of which nine and one were upregulated and downregulated, respectively, in the fat body cells established from *S. exigua* larvae after CPT treatment. Among them, seven genes (*CYP6AB14*, *CYP6AB31*, *CYP6AB60*, *CYP337B5*, *CYP6AW1*, *CYP9A27*, *CYP9A186*) belonged to the CYP3 clan and one (*CYP4G74*) belonged to CYP4. Compared to transcriptome analysis, the RT-qPCR results confirmed the expression level changes of six CYP3 clan genes *CYP6AB14*, *CYP6AB31*, *CYP6AB60*, *CYP337B5*, *CYP9A27*, *CYP9A186*, and one CYP4 clan gene *CYP4G74*. These results confirmed that members of the CYP3 and CYP4 clans are involved in the detoxification metabolism of insecticides and plant secondary metabolites in insects [19,24]. The CYP4G subfamily have been frequently associated with insecticide resistance in insects [25]. However, it has been documented that the CYP4G subfamily plays an indirect role in the insecticide resistance because they do not accept the insecticide molecule as a substrate. CYP4G enzymes are oxidative decarbonylases that catalyze the last step in cuticular hydrocarbon synthesis, making the insect cuticle thicker with a significantly reduced rate of insecticide penetration through the cuticle [26]. Subsequently, four CYP3 clan genes, *CYP9A27*, *CYP9A186*, *CYP337B5*, and *CYP321A8*, were selected to further determine the expression changes induced during CPT treatment for 10 days in S. *exigua* larvae.

During CPT treatment against *S. exigua* larvae for 10 days, it was observed that the development of *S. exigua* larvae was significantly delayed, characterized by the inhibition of weight gain and a prolonged period of larval development. In the control group, the

expression levels of *CYP9A27*, *CYP9A186*, and *CYP337B5* fluctuated significantly with the development of *S. exigua* larvae, which indicated that the expression of these genes was closely associated with the developmental period and may be involved in the regulation of the growth and development of *S. exigua* larvae. However, the expression levels of *CYP321A8* remained constant during the 10 days in the control group. In the CPT-treated group, the relative expression of *CYP9A27* was inhibited significantly at all three concentrations of CPT during the treatment period. The relative expression levels of *CYP9A186*, *CYP337B5*, and *CYP321A8* were increased or decreased with the concentrations of CPT and the treatment period. These results illustrated the potential functions of these four genes in the metabolism/detoxication or regulation of insecticides. The CYP9 family belongs to the CYP3 clan in insects and plays key roles in insecticide resistance and xenobiotic metabolism. In the lepidopteran pests *S. frugiperda*, *Helicoverpa armigera*, and *Cydia pomonella*, previous studies have demonstrated that resistance to pyrethroids is conferred by the overexpression of members of the CYP9A subfamily [26–28]. Additionally, *CYP9A1* in *Heliothis virescens* was overexpressed after exposure to thiodicarb and may play a key role in thiodicarb metabolism. In *Locusta migratoria*, the two LmCYP9A genes (*LmCYP9AQ1* and *LmCYP9A3*) were confirmed to be involved in the detoxification of different pyrethroids [28]. Specifically, *CYP9A186* confers resistance to emamectin benzoate and abamectin through enhanced metabolic detoxification in *S. exigua* [29]. *CYP321A8* has been documented to confer resistance to organophosphate (chlorpyrifos) and pyrethroid (cypermethrin and deltamethrin) insecticides in *S. exigua* [30,31]. Our findings strengthened the possibility that *CYP321A8* and *CYP9A186* are significant for the detoxification/metabolism of insecticides in *S. exigua*, while some extra experiments must be carried out. Additionally, the role of *CYP9A27* in the growth and development of *S. eixgua* provides potential new targets for pest management, and the inhibitory effects of CPT highlight the potential mechanism of CPT against insect pests.

GSTs are a diverse group of enzymes that play important roles in phase II detoxification processes. They can be further divided into seven categories, including the delta, epsilon, omega, theta, sigma, zeta, and unclassified categories [32]. In this study, a total of seventeen GST genes were identified from the transcriptomic data obtained from the fat body cells established from *S. exigua* larvae, among which ten belong to the epsilon class, two to delta, one to theta, three to sigma and one to an unclassified class. Six GST genes (*GSTd2*, *GSTe10*, *GSTe7*, *GSTe12*, *GSTe4*, and *GSTu1*) were significantly induced in the fat body cells established from *S. exigua* larvae after CPT treatment according to both DEG and RT-qPCR results. These results indicated that GSTs were associated with the detoxification of *S. exigua* to CPT, and those six GSTs might play important roles in insecticide detoxification or antioxidant protection in *S. exigua*. Previous studies have revealed that GSTs are involved in the detoxification of a variety of insecticides in insects. Specifically, the epsilon class of GSTs is involved in the metabolism of organophosphate, organochlorine, carbamate, and pyrethroid insecticides in *Chilo suppressalis*, *Anopheles funestu*, *L. migratoria*, and so on [33–35]. In this study, the relative expression levels of the two GSTs of the epsilon class, *GSTe7* and *GSTe9*, were reduced with the development of *S. exigua* larvae in the control group. However, the expression of these two GSTs, especially *GSTe7*, was induced significantly by CPT treatment, which suggested that *GSTe7* may be involved in the metabolism of CPT in *S. exigua* larvae.

## 5. Conclusions

In this study, 21 CYP and 17 GST genes were identified from the fat body cells of *S. exigua* using comparative transcriptomic analyses. The relative expression of ten CYP and seven GST genes changed significantly, suggesting their involvement in CPT metabolism. Exposure to CPT for 10 days, beginning at the second instar, resulted in delayed development of *S. exigua* larvae, as evidenced by reduced weight and extended development time. The relative expression levels of the CYP genes *CYP9A27*, *CYP9A186*, *CYP337B5* and *CYP321A8* and the GST gene *GSTe7* were significantly altered by CPT treatment compared

to the control group. These findings provide insight into the molecular-level identification of CPT metabolism/detoxification genes in *S. exigua* and demonstrate the potential use of CPT against insect pests.

**Supplementary Materials:** The following supporting information can be downloaded at https://www.mdpi.com/article/10.3390/agriculture13081494/s1: Table S1: Information for CYPs and GSTs in *S. exigua* obtained from the transcriptome data; Table S2: The corresponding protein sequences from different species for the phylogenetic analysis; Table S3: Information for primers used in RT-qPCR analysis.

**Author Contributions:** Z.Z. and L.Z. (Lan Zhang) conceived and designed the research. Z.Z. performed the experiments. Z.Z. and L.Z. (Lan Zhang) analyzed the data and wrote the manuscript. Y.Z., L.Z. (Lizhen Zhu) and H.J. reviewed the manuscript. X.L. and L.M. helped with the analysis with constructive discussions. All authors have read and agreed to the published version of the manuscript.

**Funding:** This research was supported by the Chinese National Natural Science Foundation (31672059).

**Institutional Review Board Statement:** Not applicable.

**Data Availability Statement:** The data that are presented in this study are available from the correspondence author upon request. The data are not publicly available due to privacy restrictions.

**Acknowledgments:** We would like to thank Cui for kindly providing *S. exigua* larvae.

**Conflicts of Interest:** The authors declare no conflict of interest.

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
