# Peer review of "Transcriptome-Wide Identification of Cytochrome P450s and GSTs from Spodoptera exigua Reveals Candidate Genes Involved in Camptothecin Detoxification"

_agriculture, doi:10.3390/agriculture13081494_

Round 1

Reviewer 1 Report

1.     I would suggest the authors to do correlation analysis between RNAseq and qPCR (fig 2) showing how much R2 value to support the claims.

2.     I think the concentrations could be indicated as molarity instead of g/L to help readers easily recognize the numbers.

3.     It would be great to show represntive photos of S. exigua larvae after feeding CPT as I was wondering the survival rate was not much different between control and 20 or 50 g/L groups, while the a significant reduction in the body weights.

4.     How do the authors choose the candidate genes for evaluation? Is it possible to included genes showing in the DEGs to perform qPCR upon CPT treatment? I was curious about how specific and precise of the selected transcripts.

N/A

Author Response

Responses to Reviewer 1

  1. I would suggest the authors to do correlation analysis between RNAseq and qPCR (fig 2) showing how much R2 value to support the claims.

Thanks for your good suggestions. The conclusion was drawn on the basis of the comparison with the changes of the relative expression levels of between results of RT-qPCR and the transcriptome data.

  1. I think the concentrations could be indicated as molarity instead of g/L to help readers easily recognize the numbers.

The concentrations “20 mg/L, 100 mg/L and 500 mg/L” were changed to “0.02, 0.1 and 0.5 g/L”.

  1. It would be great to show represntive photos of S. exigua larvae after feeding CPT as I was wondering the survival rate was not much different between control and 20 or 50 g/L groups, while the a significant reduction in the body weights.

Our previous studies have demonstrated that CPT inhibited strongly the growth, development and reproduction of Spodoptera exigua Hübner larvae. This figure is incited from previous studies of our group (Hongyun Jiang, 2008, Doctoral thesis, Studies on the insecticidal activity and mechanism of camptothecin and its derivatives; Jiang HY, Feng PZ, Zhang YN and Xu J, Bioactivity of three crude extracts of Camptotheca acuminato on the adults of beet armyworm (Spodoptera exigua). Plant Protection towards the 21st Century—Proc 15th Int Plant Protection Congr, Beijing, China, p. 189 (2004).

  1. How do the authors choose the candidate genes for evaluation? Is it possible to included genes showing in the DEGs to perform qPCR upon CPT treatment? I was curious about how specific and precise of the selected transcripts.

From the transcriptome data, all CYPS (21) and GSTs (17) were selected. The relative expression levels of 20 CYPs and 17 GSTs genes were verified by RT-qPCR, due to the amplification efficiency of one CPY gene is not suitable for RT-qPCR. For the candidate genes for evaluation in the role of metabolism, two reasons were considered. Firstly, the relative expression levels were changed significantly based on both results of RT-qPCR and the transcriptome data.  Secondly, it has been reported to be associated with the metabolism/detoxification of insecticides in insect pests.

Reviewer 2 Report

The manuscript by Zhenzhen Zhao et al. entitled: "Transcriptome-wide identification of cytochrome P450s and GSTs from Spodoptera exigua reveal candidate genes involved in camptothecin detoxification" reports the identification of P450- and GST-encoding genes from the beet armyworm Spodoptera exigua whose expression is significantly altered by a treatment with 10 camptothecin, an insecticide alkaloid extracted from the Happy Tree in South Cihina. These effects are further analyzed in this study. This work is rather comprehensive and quite interesting in identifying several P450 and GST genes that are significantly and specifically up- or down-regulated by the CPT treatment. Their connection with the alteration of larval development due to CPT treatment is also presented. However, I have a few comments and questions that I would like to see resolved by the authors in a revised version of this manuscript. They are presented below in the reading order and not necessarily in an order of importance.

1. line 27. It is written that camptothecin (a 20-carbon molecule) is a monoterpenoid (that are all 10-carbon molecules). This is incorrect. Camptothecin is an indole alkaloid (that’s correct) or an indolizino alkaloid (that’s more correct chemically). Please delete “monoterpenoid” in this sentence.

2. Sentence lines 31-33. We read that in humans (…) “isoiforms CYP3A1, CYP3A4 and CYP3A5 …”. This is wrong! CYP3A1 is not a human P450 but a rodent one (rat CYP3A1). This whole sentence is hard to understand, in particular the authors write that in humans P450 more closely associated with the degradation of CPTS …” Why does this “more” mean? Relatively to what? Please rewrite this sentence.

3. Sentence, lines 64-65. The word “detected” when applied to relative gene expression levels is not correct. Please change it for “monitored” or some other words similar.

4. Question about DMSO. Did you check for DMSO purity, notably the absence of oxidized DMSO contaminating chemicals frequently found in commercial sources of DMSO? These oxidized DMSO derivatives are known to be highly deleterious to P450 activity (See: Scaduto RC. 1995, Free Radic Biol Med 18(2), 271-277).

5. Line 159. It is written that “1 gene, CYP333B4, was downregulated significantly”, but these are not 1 but 11 P450 genes that are indicated with a blue star (downregulated significantly) in Fig. 1. Could the authors resolve this apparent discrepancy?

6.In Fig. 2a, both CYP6AW1 and CYP333A12 appears to be not affected by CPT treatment for both RT-qPCR and DGE indicators. But these two genes are indicated as significantly upregulated in Fig. 1. Could the authors resolve this intriguing discrepancy?

7. Line 214 and Fig. 4 legend. In line 214, we read that the highest camptothecin concentration used by the authors is 500 mg/L, while in the legend of Fig. 4 we read that it is 1000 mg/L. Which one is correct?

8. Figure 4. I suggest to the authors to indicate at the top of each panel which is what (no CPT, CPT 20 mg/L etc.). That would help the general reader.

9. Line 233. It is written that the relative expression of the CYP9A27 gene was reduced significantly by CPT. This is (again see my comment 6) this is totally opposite to what is presented in Fig. 1, where CYP9A27 is highlighted with a green star meaning it is significantly upregulated by CPT treatment.

10. Line 229. We read: “However, the relative expression of CYP321A8 remained unchanged throughout the tested period…” I totally disagree. At day 1 und for the three CPT concentrations tested, Fig. 5 shows that the expression for this gene is systematically half the level observed in the control experiment. This is clearly not what I call “unchanged”. The authors should correct this.

11. Could the authors explain what was the rationale they used to select the 4 CYP genes they present in more details from line 231?

12. Question. Have we any information on the chemical nature(s) of the metabolite(s) produced by P450 activity(ies) on camptothecin? If yes, that would be, to my opinion, informative for the general reader to add a scheme illustrating this in the introduction of the revised version of this manuscript.

13. Question. Last but not least, have we any information on the activities catalyzed by the upregulated P450s identified in this interesting study?

14. Comment. Among the upregulated P450 in S. exigua, the authors have identified CYP4G74. It is well known that enzymes of the CYP4G subfamily are frequently associated with insecticide resistance in insects but with a rather indirect role. They do not accept the insecticide molecule as a substrate. Rather, CYP4G enzymes are oxidative decarbonylases that catalyze the last step in cuticular hydrocarbon synthesis, making the insect cuticle thicker with a significantly reduced rate of insecticide penetration through the cuticle (See: Feyereisen R. 2020. Mol Phylogenet Evol 143, e106695). Could the authors comment on that in their discussion in the revised version of this manuscript? Because, they insist in their text on P450s implied in the direct resistance by modifying biochemically the camptothecin molecule itself. This very well could not be the case for all upregulated P450 they have identified.

Minor

1. The title is: “Transcriptome-wide Identification of Cytochrome P450s and 2 GSTs from Spodoptera exigua Reveal Candidate Genes Involved in Camptothecin Detoxification" and it contains a grammar error. The verb "reveal" agrees with "transcriptome-wide identification, hence it should read: “reveals”. Please correct this.

2. Line 31. Instead of “in Homo species” that is a quite unusual way of speaking, please write: “in humans,”.

3. Lines 191-192. A space is missing between these two lines.

4. Line 195. We read: “… are shown in Fig. 3 and Fig. 4, respectively”. Please change the “in” for an “on”.

5. Sentence. Line 264-267. A “respectively” is missing at line 266 after the word “downregulated”.

6. Lines 283 and 286. The authors use two “however” in three sentences; That is too much. Please change this.

Please refer to my minor comments 1-6.

P. Urban

Author Response

Responses to Reviewer 2

The manuscript by Zhenzhen Zhao et al. entitled: "Transcriptome-wide identification of cytochrome P450s and GSTs from Spodoptera exigua reveal candidate genes involved in camptothecin detoxification" reports the identification of P450- and GST-encoding genes from the beet armyworm Spodoptera exigua whose expression is significantly altered by a treatment with 10 camptothecin, an insecticide alkaloid extracted from the Happy Tree in South China. These effects are further analyzed in this study. This work is rather comprehensive and quite interesting in identifying several P450 and GST genes that are significantly and specifically up- or down-regulated by the CPT treatment. Their connection with the alteration of larval development due to CPT treatment is also presented. However, I have a few comments and questions that I would like to see resolved by the authors in a revised version of this manuscript. They are presented below in the reading order and not necessarily in an order of importance.

Thank you for your kindly comments.

  1. line 27. It is written that camptothecin (a 20-carbon molecule) is a monoterpenoid (that are all 10-carbon molecules). This is incorrect. Camptothecin is an indole alkaloid (that’s correct) or an indolizino alkaloid (that’s more correct chemically). Please delete “monoterpenoid” in this sentence.

Thanks. “monoterpenoid” was deleted.

  1. Sentence lines 31-33. We read that in humans (…) “isoiforms CYP3A1, CYP3A4 and CYP3A5 …”. This is wrong! CYP3A1 is not a human P450 but a rodent one (rat CYP3A1). This whole sentence is hard to understand, in particular the authors write that in humans P450 more closely associated with the degradation of CPTS …” Why does this “more” mean? Relatively to what? Please rewrite this sentence.

 Thanks. This sentence was changed to “In humans, cytochrome P450 monooxygenases (P450s or CYPs) have been documented to involve in the degradation of CPTs, especially P540 isoforms 1A1, 3A4 and 3A5 (CYP1A1, CYP3A4, CYP3A5) [1-3]”.

  1. Sentence, lines 64-65. The word “detected” when applied to relative gene expression levels is not correct. Please change it for “monitored” or some other words similar.

Thanks. Lines 64-65, “detected” was changed to “monitored”.

  1. Question about DMSO. Did you check for DMSO purity, notably the absence of oxidized DMSO contaminating chemicals frequently found in commercial sources of DMSO? These oxidized DMSO derivatives are known to be highly deleterious to P450 activity (See: Scaduto RC. 1995, Free Radic Biol Med 18(2), 271-277).

Thanks. The purity of DMSO was added Line 70. Additionally, the concentration of DMSO is lower than 0.1% in the test solutions.

  1. Line 159. It is written that “1 gene, CYP333B4, was downregulated significantly”, but these are not 1 but 11 P450 genes that are indicated with a blue star (downregulated significantly) in Fig. 1. Could the authors resolve this apparent discrepancy?

Thanks. The stars with different colors marked before CYP or GST proteins of S. exigua were significantly upregulated (green stars), significantly downregulated (red stars) or unchanged (blue stars) according to the transcriptome data (|Log2 FC|≥1, P<0.05).

6.In Fig. 2a, both CYP6AW1 and CYP333A12 appears to be not affected by CPT treatment for both RT-qPCR and DGE indicators. But these two genes are indicated as significantly upregulated in Fig. 1. Could the authors resolve this intriguing discrepancy?

According to transcriptome data, the relative expression levels of both CYP6AW1 (Log2 FC=1.1907, P<0.05) and CYP333A12 (Log2 FC=1.5775, P<0.05) in the Fig.1 and Fig.2 were up-regulated significantly based on the (|Log2 FC|≥1, P<0.05).  However, the relative expression levels of CYP6AW1 (Log2 FC=1.2822, P> 0.05) and CYP333A12 (Log2 FC=1.2564, P> 0.05) based on the Student's t test performed with the SPSS 26.0 Software Package (SPSS Inc.; Chicago, IL, USA).

  1. Line 214 and Fig. 4 legend. In line 214, we read that the highest camptothecin concentration used by the authors is 500 mg/L, while in the legend of Fig. 4 we read that it is 1000 mg/L. Which one is correct?

        Thanks for your suggestions. Fig. 4 legend, “……and 1000 mg/L (d)” was changed to “…….and 0.5 g/L (d)”.

  1. Figure 4. I suggest to the authors to indicate at the top of each panel which is what (no CPT, CPT 20 mg/L etc.). That would help the general reader.

Thanks for your suggestions. “CK, 0.02 g/L, 0.1 g/L, and 0.5 g/L” were added for the Fig. 4a, b, c and d, respectively.

  1. Line 233. It is written that the relative expression of the CYP9A27 gene was reduced significantly by CPT. This is (again see my comment 6) this is totally opposite to what is presented in Fig. 1, where CYP9A27 is highlighted with a green star meaning it is significantly upregulated by CPT treatment.

Thanks for your suggestions. The relative expression of CYP9A27 gene was reduced significantly by CPT during the development the S. exigua larvae in the Fig. 5. However, in the Fig. 1, the relative expression of CYP9A27 gene was up-regulated significantly in the fat body cells of S. exigua. I think the opposite results are due to the whole body of S. exigua larvae used in Fig. 5 and only fat body cells used in Fig. 1.  In order to stained clearly these, “3.1 Identification of candidate P450s and GSTs” was changed to “Identification of candidate P450s and GSTs from the fat body cells of S. exigua”; “3.2 Real-time quantitative PCR validation of candidate genes” was changed to “Real-time quantitative PCR validation of candidate genes from the fat body cells of S. exigua”; “3.3 Effects of CPT on the development of S. exigua” was changed to “3.3 Effects of CPT on the development of S. exigua larvae”; and “3.4 Effect of CPT on the expression of CYP and GST genes during the development of S. exigua” was changed to “3.4 Effect of CPT on the expression of CYP and GST genes during the development of S. exigua larvae ”.

  1. Line 229. We read: “However, the relative expression of CYP321A8 remained unchanged throughout the tested period…” I totally disagree. At day 1 und for the three CPT concentrations tested, Fig. 5 shows that the expression for this gene is systematically half the level observed in the control experiment. This is clearly not what I call “unchanged”. The authors should correct this.

Thanks for your suggestions. This sentence was changed to “However, the relative expression of CYP321A8 remained unchanged throughout the tested period of S. exigua larval development in the control group”.

  1. Could the authors explain what was the rationale they used to select the 4 CYP genes they present in more details from line 231?

Thanks for your suggestions. Sentences “To study the effect of CPT on the expression of CYP and GST genes, four CYP3 clan genes (CYP9A27, CYP9A186, CYP337B5 and CYP321A8) and two GSTs (GSTe8 and GSTs3) genes were selected to conduct further studies during the development of S. exigua larvae based on both results of transcriptome analysis and RT-qPCR verification” were added.

  1. Question. Have we any information on the chemical nature(s) of the metabolite(s) produced by P450 activity(ies) on camptothecin? If yes, that would be, to my opinion, informative for the general reader to add a scheme illustrating this in the introduction of the revised version of this manuscript.

Thanks for your questions and suggestions. There are few reports about the chemical nature(s) of the metabolite(s) produced by P450 activity(ies) on camptothecin.

  1. Question. Last but not least, have we any information on the activities catalyzed by the upregulated P450s identified in this interesting study?

Thanks for your questions. In this study, the candidate genes might involve in camptothecin detoxification/metabolism were found and identified, which will provide a basis information about CPT metabolism/detoxification at the molecular level and demonstrate the potential use of CPT against insect pests. In the future study, it will conduct a systematic work to study the activities of these identified genes which might involve in the camptothecin detoxification/metabolism.

  1. Comment. Among the upregulated P450 in S. exigua, the authors have identified CYP4G74. It is well known that enzymes of the CYP4G subfamily are frequently associated with insecticide resistance in insects but with a rather indirect role. They do not accept the insecticide molecule as a substrate. Rather, CYP4G enzymes are oxidative decarbonylases that catalyze the last step in cuticular hydrocarbon synthesis, making the insect cuticle thicker with a significantly reduced rate of insecticide penetration through the cuticle (See: Feyereisen R. 2020. Mol Phylogenet Evol 143, e106695). Could the authors comment on that in their discussion in the revised version of this manuscript? Because, they insist in their text on P450s implied in the direct resistance by modifying biochemically the camptothecin molecule itself. This very well could not be the case for all upregulated P450 they have identified.

Thanks for your suggestions. The comments “It has been reported that the CYP4G subfamily are frequently associated with insecticide resistance in insects with an indirect role due to they do not accept the insecticide molecule as a substrate. CYP4G enzymes are oxidative decarbonylases that catalyze the last step in cuticular hydrocarbon synthesis, making the insect cuticle thicker with a significantly reduced rate of insecticide penetration through the cuticle [26].” were added in line 275.

Minor

  1. The title is: “Transcriptome-wide Identification of Cytochrome P450s and 2 GSTs from Spodoptera exigua Reveal Candidate Genes Involved in Camptothecin Detoxification" and it contains a grammar error. The verb "reveal" agrees with "transcriptome-wide identification, hence it should read: “reveals”. Please correct this.

The title was changed to “Transcriptome-wide Identification of Cytochrome P450s and GSTs from Spodoptera exigua Reveals Candidate Genes Involved in Camptothecin Detoxification”.

  1. Line 31. Instead of “in homo species” that is a quite unusual way of speaking, please write: “in humans,”.

“in Homo species” was changed to “in humans”.

  1. Lines 191-192. A space is missing between these two lines.

A space was added between line 191 and 192.

  1. Line 195. We read: “… are shown in Fig. 3 and Fig. 4, respectively”. Please change the “in” for an “on”.

Line 195, “… are shown in Fig. 3 and Fig. 4, respectively” was changed to “… are shown on Fig. 3 and Fig. 4, respectively”.

  1. Sentence. Line 264-267. A “respectively” is missing at line 266 after the word “downregulated”.

Line 266, “respectively” was added after the word “downregulated”.

  1. Lines 283 and 286. The authors use two “however” in three sentences; That is too much. Please change this.

Sentence “However, the relative expression levels of CYP9A186……….” was change to “the relative expression levels of CYP9A186………”.

Round 2

Reviewer 1 Report

The authors have addressed all my comments.

Author Response

Thank you.